# Keratin-Based Composite Bioactive Films and Their Preservative Effects on Cherry Tomato

**DOI:** 10.3390/molecules27196331

**Published:** 2022-09-26

**Authors:** Lanlan Wei, Shuaijie Zhu, Huan Yang, Zhiheng Liao, Zexuan Gong, Wenwen Zhao, Yan Li, Jinyan Gu, Zhaohui Wei, Jianting Yang

**Affiliations:** 1College of Food Engineering, Anhui Science and Technology University, Chuzhou 233100, China; 2Anhui Jinhuacui Food Co., Ltd., Chuzhou 239200, China

**Keywords:** pig nail keratin, keratin/gelatin/glycerin/curcumin film, preservation, cherry tomatoes

## Abstract

In this study, keratins were extracted from pig nail waste through the reduction method using L-cysteine as a reductant. Curcumin was successively incorporated in a mixed solution including keratin, gelatin, and glycerin to prepare different kinds of keratin/gelatin/glycerin/curcumin composite films. The morphology of the keratin/ gelatin/glycerin/curcumin composite films were examined using scanning electron microscopy. The structures and the molecular interactions between curcumin, keratin, and pectin were examined using Fourier transform infrared spectroscopy and X-ray diffraction, and the thermal properties were determined through thermogravimetric analysis. The tensile strengths of keratin/gelatin/glycerin/curcumin and keratin/gelatin/curcumin composite films are 13.73 and 12.45 MPa, respectively, and their respective elongations at break are 56.7% and 4.6%. In addition, compared with the control group (no film wrapped on the surface of tomato), the ratio of weight loss of the keratin (7.0%)/gelatin (10%)/glycerin (2.0%)/curcumin (1.0%) experimental groups is 8.76 ± 0.2%, and the hardness value of the tomatoes wrapped with composite films is 11.2 ± 0.39 kg/cm^3^. Finally, the composite films have a superior antibacterial effect against *Staphylococcus aureus* and *Escherichia coli* because of the addition of curcumin. As the concentration of curcumin reaches 1.0%, the antibacterial activity effect of the film is significantly improved. The diameter of the inhibition zone of *E. coli* is (12.16 ± 0.53) mm, and that of *S. aureus* is (14.532 ± 0.97) mm. The multifunctional keratin/gelatin/glycerin/curcumin bioactive films have great potential application in the food packaging industry.

## 1. Introduction

Cherry tomatoes are a source of nutrients and other healthy minerals, including lycopene, phenolic compounds, and vitamin C, that benefit the human body [1]. However, similar to other fruits and vegetables, tomatoes are also sensitive to storage environment, and rot easily during transportation and the market process [2,3]. The weight, firmness, and color of cherry tomatoes are essential quality indicators; these quality indicators determine consumer acceptance and market success [4]. Food packaging plays a vital role in maintaining food quality and senescence properties. Additionally, food packaging ensures food safety and extends food shelf life [5]. The packaging industry uses a large amount of non-degradable petroleum-based synthetic plastics annually to mechanically, chemically, and microbiologically protect food products [6]. The increasing production and usage of these non-degradable plastic packaging can cause environmental problems [7]. Thus, food packaging researchers have focused on biodegradable alternatives, such as biopolymers: carbohydrates, proteins, and lipids [8]. Film packaging with edible, biodegradable, and antioxidant properties is an innovative packaging technology and a development trend.

Keratin is abundant in nature and a biopolymer material widely used in various fields; it is preferred over other natural polymers owing to its biocompatibility and biodegradability [9,10,11]. In addition, keratin is a structural fibrous protein, which is the main component of wool, feathers, nails, horns, and other epithelial coverings [12]. Studies show that the performance of keratin films can be improved by combining keratin with natural [13] and synthetic polymers [14]. Pig nails are rich in protein and amino acids. However, about a dozen tons of pig nail waste are currently generated and abandoned annually due to their limited application, resulting in a severe waste of resources. Thus, the reasonable conversion of waste pig nails into useful biopolymer material can expand its application, and meet the development trend of green environmental protection.

Gelatin can be obtained from the partial hydrolysis of collagen, and it is widely used in the food industry for coating fudge, jelly, and other edible items, owing to its good biocompatibility, biodegradability, and film-forming properties [15,16]. Several studies show that cracks may appear on thin films synthesized via the sol–gel method, which can adversely affect the response of any device; thus, cracks in thin films are undesirable [17]. Glycerin can be used during the preparation of the precursor solution to prevent crack formation [18]. However, the modification of gelatin film with natural active substances can effectively improve its antioxidant activity. Curcumin (Cur) is one of the major bioactive compounds of curcuma rhizome, and it is used in medical or pharmacological applications due to its antioxidant, antibacterial, fall hematic fat, antitumor, anti-inflammatory, cholagogic, anticancer, and wound-healing effects [19,20,21,22]. Curcumin is often used as a coloring agent for food packages. Generally, the bioactivities of curcumin make it ideal for food and medical applications.

Therefore, this study attempted to produce a modified biodegradable packaging film through a combination of gelatin, keratin, glycerol, and curcumin for the first time, and their effects on film were investigated. In addition, the effects of curcumin loading on the physicochemical properties of cherry tomatoes and the growth of *S. aureus* (*staphylococcus aureus*) and *E. coli* (*Escherichia coli*) were also evaluated.

## 2. Results and Discussion

### 2.1. SEM of Pig Nail Powders and Pig Nail Keratin

To intuitively understand the morphological changes in pig nail powders and pig nail keratin, they were analyzed via SEM. The result show that the external structure of the undissolved pig nail powder is neatly arranged (Figure 1a,b). During the process of dissolution, the overall morphological structure of pig nail powder changes with shedding, due to the destruction of the intermolecular force and chemical bonds (Figure 1c,d).

### 2.2. XRD Analysis of Pig Nail Powders and Pig Nail Keratin

The diffraction peaks of 2θ = 9° and 21° are related to the structure of α-helix and β-sheet for keratin, respectively. If there is no, or weak, pretreatment for pig nail powder, it has its own crystal region, as shown in Figure 2. However, the characteristic peak of pig nail keratin at 9° is significantly weaker than that of pig nail powders, indicating the destruction of the α-helical structure of pig nail powders because of the dissolution procedure. In addition, the characteristic peak of pig nail keratin is stronger than that of pig nail powders at 21°, indicating an increase in the content of β-sheet structure [23].

### 2.3. SEM of the Composite Films

SEM images provide detailed information on the surface and cross-section of the composite films. SEM results show that the surface of 10% pure gelatin film is flat and smooth (Figure 3a). When the concentration of keratin is 4%, little granules are observed on the surface of the composite film (Figure 3b). However, with the increasing keratin content in the sample, the surface of the keratin/gelatin film becomes rough. When the concentration of keratin is 7%, some granules are observed on the surfaces of the composite (Figure 3c). Nevertheless, the surface of the 10% keratin composite film is uneven and extremely rough (Figure 3d). This phenomenon could be attributed to the fact that keratin has a crystal structure and gelatin has no crystal structure, thus, leading to the phase separation when mixed with a high concentration of keratin. When the content of keratin is 7%, the surface of the keratin/gelatin/glycerol film is smooth and tight, with superior uniformity. Hence, the keratin (7%)/gelatin (10%)/glycerin (2.0%)/curcumin (1.0%) composite film was selected for follow-up study. Additionally, as shown in Figure 3e,f, few pores are observed in the cross-section of the composite surface. The cross-section of the composite surface is also fully cured and possesses a compact structure.

### 2.4. Mechanical Property of the Composite Films

Mechanical properties are an important factor used to determine the durability of edible packaging films [24]. There is no significant difference in tensile strength between keratin (7%)/gelatin (10%)/curcumin (1.0%) (12.45 Mpa) and keratin (7%)/gelatin (10%)/glycerin (2.0%)/curcumin (1.0%) composite films (13.73 Mpa). In addition, the elongation at break of the keratin (7%)/gelatin (10%)/glycerin (2.0%)/curcumin (1.0%) composite film is 56.7%, which is higher than that of the keratin (7%)/gelatin (10%)/curcumin (1.0%) composite films (Figure 4). These phenomena can be explained as follows: when glycerol is introduced to keratin and gelatin, the hydrogen bond interaction between the keratin, gelatin, and glycerol hinders the movement and rotation of the molecular chain, resulting in a highly rigid molecular chain [25].

### 2.5. Characterization of Different Kinds of Composite Films

The FTIR spectra of keratin/gelatin/glycerin/curcumin, keratin/gelatin/curcumin, and keratin are shown in Figure 5A. A special absorption peak of peptide bonds (–CONH–) is observed, attributed to the characteristics of protein in pig nail keratin and gelatin. The amide I bands are observed at 1647 cm^−1^, representing C=O stretching vibration. The amide II bands observed at 1548 cm^−1^ represent N-H bending vibrations. Moreover, with the addition of gelatin, the absorption peak of –CONH– is considerably enhanced. Except in Figure 5Aa, no considerable absorption peaks of O–H and C–H are observed. However, Figure 5Ab–Ad show broadband at 3286 and 2921 cm^−1^, representing the stretching vibrations of O–H and C–H of methylene in the methyl group of the gelatin and glycerin. The peak strength of O–H and C–H increases with the addition of glycerin. The above results indicate that the combination of keratin, gelatin, glycerin, and curcumin does not produce new chemical bonds, but completely exists in keratin/gelatin/glycerin/curcumin composite films.

The XRD pattern obtained for pure pig nail keratin exhibits broad diffraction peaks at 2θ = 9° and 20°, which are typical fingerprints of partially crystalline materials. After the addition of gelatin to the sample, no diffraction peak is observed at 2θ = 9°, indicating that the overall crystallinity of the blend films is slightly lower than the crystallinity of the gelatin, due to the addition of keratin, which contains a small crystalline structure and a large amorphous region. As shown in Figure 5B, the appearance of a serial of 2θ diffraction angles at 14° confirms the crystalline structure of curcumin [26].

The thermal properties of the films are shown in Figure 5C. Just for the keratin/gel/gly/cur film, the keratin films show a three-step thermal degradation pattern, as observed from the TGA curve. The initial degradation step associated with water loss occurs between 25 °C and 130 °C, and the second degradation step is observed in the temperature range of 130–400 °C, with a maximum degradation rate obtained at 365 °C, which is due to the degradation of keratin chain backbone. The third step of degradation occurs between 400 and 800 °C due to the intramolecular hydrogen bond in the keratin chain structure. As shown in Figure 5C, the weight loss of the keratin/gel/gly/cur film is higher than that of pig nail keratin, ranging from 146–310 °C. Moreover, the keratin/gel/gly/cur film shows higher thermal stabilities than the pure pig nail keratin, which is in the range of 310–800 °C. The results show that the addition of gelatin and glycerin could provide effective reinforcement against the thermal degradation of the pig nail keratin matrix, increasing the thermal stability of the matrix.

### 2.6. The Application to Preserve Cherry Tomatoes

In this study, the fruits were stored at room temperature without a packing box. Figure 6A shows the appearance changes in tomatoes packaged with the keratin/gel/gly/cur composite films with different concentrations of curcumin after 10 days of storage. For the control group, which is not film packaged, the skins of the tomatoes are considerably wrinkled. Soft sarcocarp without shape is observed in the cross-section of the tomatoes. Moreover, the skin color of the tomatoes is dim. However, the tomatoes packaged with the keratin/gel/gly/cur composite films after 10 days of storage are less wrinkled, and also have a bright red color and smooth appearance. Moreover, among all the experimental groups, the tomatoes packaged with the keratin/gel/gly/cur (1.0%) composite film have better preservation. These results are similar to those reported in the literature [27].

As shown in Figure 6B, the ratio of weight loss the control group (tomatoes without films) and experiment group shows an increasing trend. However, the ratio of weight loss for cherry tomatoes wrapped up with the keratin/gel/gly/cur composite films is lower than that of the control groups. Moreover, the ratio of weight loss (11.98 ± 0.43%) of cherry tomatoes packaged with the keratin/gel/gly/cur (1.0%) composite film group is lower than that of other composite films. Cherry tomatoes packaged with composite films are protected from water and oxygen permeabilities, reducing respiration, transpiration, water loss, and organic matter consumption [28].

Hardness can be used to determine the texture changes in fruits and vegetables during storage [29]. As shown in Figure 6C, the hardness of the control and experiment groups display a decreasing trend during the storage at room temperature, irrespective of the treatment of composite films. However, the hardness value of cherry tomatoes packaged with the keratin/gel/gly/cur (1.0%) composite film is 3.25 ± 0.43 kg/cm^3^, higher than that of other composite films. In addition, the thickness is important to evaluate the mechanical properties. The thickness of the keratin/gel/gly/cur (1.0%) composite film is about 0.10 ± 0.03 mm.

Changes in color coordinates of cherry tomatoes during storage are summarized in Table 1. The values of luminosity (L*), chroma (C*ab), and hue (H*ab) decrease after 10 days. The results indicate that the color of tomatoes becomes redder and less vivid due to tissue softening and skin browning. Compared with the control group, the decreases in values of luminosity (L*), chroma (C*ab), and hue (H*ab) are not very pronounced in tomatoes packaged with films (Table 1). The reason for this decreased value is that the process of browning caused by enzymatic and non-enzymatic reactions is delayed through the application of packaging [30].

### 2.7. The Antibacterial Activity of Keratin/gel/gly/cur Composite Films

*Escherichia coli* and *Staphylococcus aureus* were taken as the representative strain and the diameter of the inhibition zone was treated as the evaluation indicator to determine the influence of different concentrations of curcumin on the antibacterial activity of the keratin/gel/gly/cur composite films. As shown in Figure 6D, with the increased curcumin concentration, the diameter of the inhibition zone of composite films gradually expands. When the concentration of curcumin reaches 1.0%, the antibacterial activity effect of the film is significantly improved. The diameter of the inhibition zone of *E. coli* is (12.16 ± 0.53) mm, and that of *S. aureus* is (14.532 ± 0.97) mm.

## 3. Materials and Methods

### 3.1. Materials and Reagents

Analytical grade chemicals and reagents were used. Urea (CH_4_N_2_O), L-cysteine (C_3_H_7_NO_2_S, molecular mass: 121.16 Dalton), sodium hydroxide (NaOH), hexafluoroisopropanol (C_3_H_2_F_6_O), gelatin (gel, gel strength ~250 Bloom), and glycerol (gly, molecular mass: 92.09) were purchased from Shanghai Macklin Biochemical company (Shanghai, China, www. macklin.cn.qianyan.biz (accessed date on 8 September 2021)). Hydrochloric acid (HCl), barium chloride (BaCl_2_), oxalic acid (H_2_C_2_O_4_), curcumin (cur, molecular mass: 368.39), and phenolphthalein (C_20_H_14_O_4_) were purchased from Sinopharm Chemical Reagent Co., Ltd. (Shanghai, China, www.reagen.com.cn (accessed date on 10 September 2021)). Plate count agar (PCA) and nutrient broth (NB) were purchased from Shanghai Bio-way Technology Co., Ltd. (Shanghai, China, www.bw-bio.com (accessed date on 15 September 2021)).

### 3.2. Extraction of Keratin from Pig Nails

The pig nail keratin was extracted from clean pig nails following the procedure reported in the previous study [31], with appropriate adjustments. As shown in Figure 7, 3.0 g of pig nails were initially incubated in 4 mol/L urea, and 0.45 g of L-cysteine was added to the mixture. Then, the aqueous solution was stirred at 70 °C and 150 rpm for 16 h. After the supernatant was collected, the solution was centrifuged at 10,000 rpm for 15 min. The pH of the supernatant solution was adjusted to 4.0 using 0.2 mol/L of HCl. Then, the keratin was precipitated and washed with deionized water and freeze-dried with a vacuum freeze dryer to obtain pig nail keratin powders.

### 3.3. Preparation of Keratin/Gelatin/Glycerol/Curcumin Films

The obtained pig nail keratin powders were dissolved in hexafluoroisopropanol at room temperature with the concentration of 4%, 7.0%, 10% (*w*/*v*) to form keratin solution. Then, 1.0 g of gelatin was weighed and added to the keratin solution, and the mixed solution was stirred at 50 °C until the gelatin was completely dissolved; a light yellow and transparent solution was formed. After the obtained solution was cooled to room temperature, 0.2 g of glycerol was added and stirred evenly at room temperature. Then, curcumin powders were added to the mixed solution, so that the concentrations of curcumin were 0.4%, 0.6%, 0.8%, 1.0%, 1.2% (*w*/*v*), and stirred ultrasonically for 5 min. Then, the mixed solution was removed, about 5.0 mL poured into a circular mold with a diameter of 8 cm, and placed in a drying oven (35 °C, 24 h) to obtain the keratin/gel/gly/cur composite films.

### 3.4. Characterization

The surface morphology of the composite films was investigated using a scanning electron microscope (SEM, Supra55, Zeiss, German) at 20 kV. The functional groups of the nanofiber membrane were confirmed via ATR-IR spectroscopy (FTIR, NicoletiS10, Thermo Fisher, USA), performed in a range of 4000–1000 cm^−1^. The X-ray diffraction analysis was performed using an X-ray diffractometer (XRD, XD-3X, Persee general, China), and the XRD spectra were recorded in the 2θ range of 5°–60° at a step of 0.02°. Thermal stability was observed using a thermogravimetric analyzer (TGA, TG 209 F3, NETZSCH, Germany) under the continuous nitrogen flow (20 mL/min) at a temperature range of 25–800 °C and a 15 °C/min rate.

### 3.5. Mechanical Properties

The tensile strength and elongation at break of the different composite films were tested using a tensile testing machine (WDT-10, FuSide Instruments and Equipment Co., Ltd., Wuxi, China) under the natural condition with initial grips separation at 50 mm and probe speed of 30 mm/min. The tensile strength was the maximum tensile stress when the membrane broke and the data were read directly from the device. The equation of calculating elongation at break followed:The elongation at break = (L_0_ − L)/L × 100%,
where L represents the initial length of fixture and L_0_ represents the length of fixture after the film breaks.

### 3.6. Application of Composite Films to Store Cherry Tomatoes

The cherry tomatoes were purchased from a local supermarket and washed in the laboratory with tap water. After the cherry tomatoes were dried under natural conditions, they were wrapped up with the prepared composite films in random groups. Then, the fruits of control group (no film wrapped on the tomato surface), and experiment group (keratin/gelatin/glycerin/curcumin composite films with the different concentrations of curcumin wrapped on the tomato surface) were placed at room temperature. The quality indicators of tomatoes from control group and experiment group, including the ratio of weight loss, hardness, and surface color, were determined every two days.

### 3.7. Antibacterial Property of Composite Films

The different composite films at different concentrations of curcumin were cut into circular sheets of 5 mm diameter and irradiated under ultraviolet lamp for 30 min. *Escherichia coli* and *Staphylococcus aureus* strains were inoculated into a 20 mL NB medium, then cultivated at 38 °C for 24 h on a constant temperature shaker. Afterward, 60 μL of two kinds of bacterial fluid dilution were coated on the PCA plates, the composite films were placed on plates, and the *E. coli* plate and *S. aureus* plate were cultured in an incubator at 38 °C constant temperature for 24 h.

### 3.8. The Ratio of Weight Loss

The weight loss of cherry tomatoes could be attributed to the respiration and moisture evaporation during storage. The ratio of weight loss for cherry tomatoes packaged and unpackaged with the composite films before and after the storage was calculated using the equation below.
The ratio of weight loss = (M_1_ − M_2_/M_1_) × 100%, 
where M_1_ represents the initial weight of the cherry tomatoes before storage, and M_2_ represents the weight of the cherry tomatoes at different storage times.

### 3.9. Hardness

Hardness is the pressure (N) of the force-measuring spring per unit area (S) of one cherry tomato. When the hardness unit was kg/cm^3^, the depth of depression was stochastic. In this study, the TA41 cylindrical probe was selected and installed, and the distance between the probe and the base platform was corrected. In addition, the compression speed and the degree of compression were 1.0 mm/s and 50%, respectively. The hardness was measured using a hardness tester (CT3-TextureAnalyzer, Brookfield Co., Ltd., Middleborough, MA, USA). The probe selection is applicable to fruit and vegetables.

### 3.10. Thickness

Thickness was measured with vernier caliper (Shanghai Shen Gong Measuring Tool Co., Ltd., China). The film was folded in half three times, measured with vernier caliper (0.02mm), and then calculated using the equation below.
The thickness = L/8,
where L represents the thickness of film with 8 layers.

### 3.11. Surface Color

The surface color of cherry tomatoes was treated as one of the quality parameters and determined using a colorimeter (NR110, 3nh Global, Shenzhen, China). The samples packaged with different composite films of different curcumin concentrations were stored at room temperature, and the values of lightness (L*), redness (a*), and yellowness (b*) were periodically evaluated. Each sample was measured in parallel three times for statistical analysis.

### 3.12. Statistical Analysis

The experimental errors were solved by recording triplicate values and the results were presented with a mean ± standard deviation. Significant differences among the samples were analyzed by ANOVA with SPSS statistics 26. A significant level was set at *p* < 0.05.

## 4. Conclusions

In this study, different kinds of keratin/gelatin/glycerin/curcumin composite films are prepared, and the novel keratin (7.0%)/ gelatin (10%)/glycerin (2.0%)/curcumin (1.0%) composite film has the best effect on the preservation of the tomatoes. Uniform distribution of curcumin added to keratin/gel/gly is observed using SEM, and their interactions are confirmed though FTIR, XRD, and TGA spectra. In addition, the composite films have a superior antibacterial effect against *Staphylococcus aureus* and *Escherichia coli* because of the addition of curcumin. Moreover, the ratio of weight loss for cherry tomatoes wrapped up with the keratin/gel/gly/cur composite films is lower than that of the control groups (tomatoes without films), the hardness value of cherry tomatoes wrapped up with keratin/gel/gly/cur (1.0%) composite film is 3.25 ± 0.43 kg/cm^3^ and higher than that of the other composite films, at least compared with the control group, while the decreases in the values of luminosity (L*), chroma (C*ab), and hue (H*ab) are not very pronounced in tomatoes with films. Therefore, multifunctional keratin/gel/gly/cur films possess great potential use in active fruit and vegetable packaging.

## Figures and Tables

**Figure 1 molecules-27-06331-f001:**
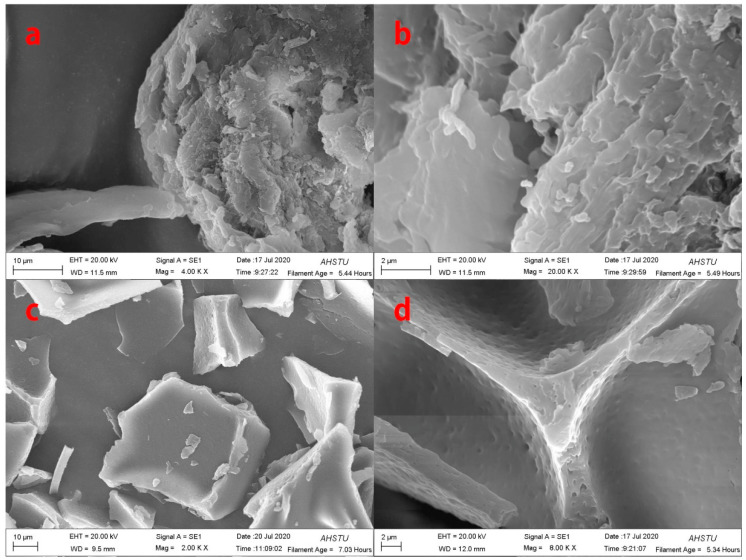
SEM images of: (**a**,**b**) pig nail powders; (**c**,**d**) pig nail keratin.

**Figure 2 molecules-27-06331-f002:**
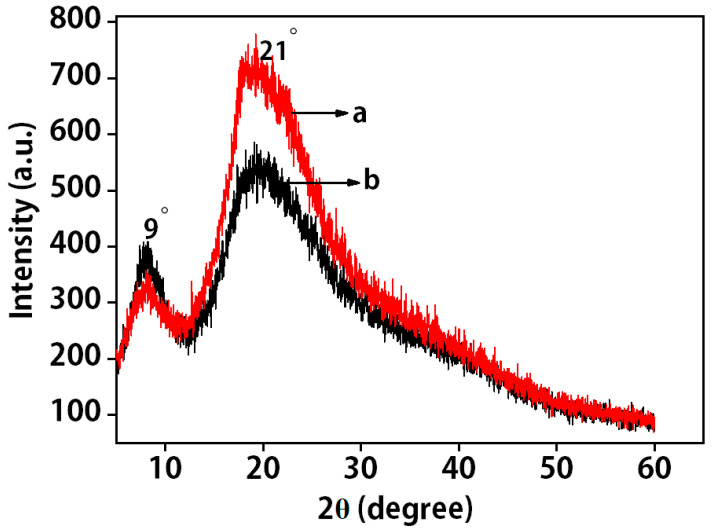
XRD analysis of (**a**) pig nail keratin; (**b**) pig nail powders.

**Figure 3 molecules-27-06331-f003:**
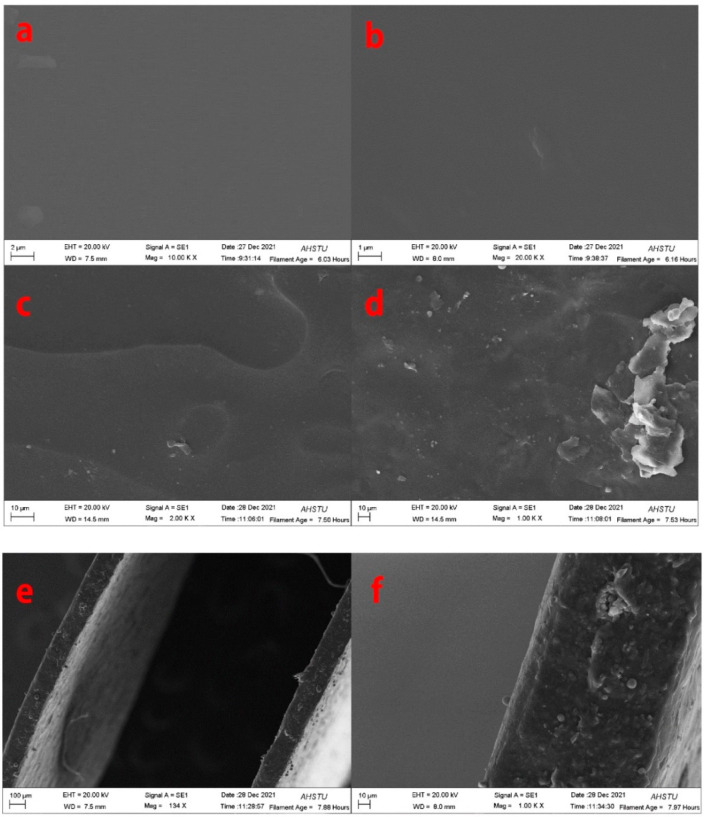
SEM images of surfaces for (**a**) pure gelatin film, (**b**) keratin (4%)/gelatin (10%)/glycerin (2.0%)/curcumin (1.0%) composite film, (**c**) keratin (7%)/gelatin (10%)/glycerin (2.0%)/curcumin (1.0%) composite film, and (**d**) keratin (10%)/gelatin (10%)/glycerin (2.0%)/curcumin (1.0%) composite film. The SEM images of the cross-section of (**e**,**f**) keratin (7%)/gelatin (10%)/glycerin (2.0%)/curcumin (1.0%) composite film.

**Figure 4 molecules-27-06331-f004:**
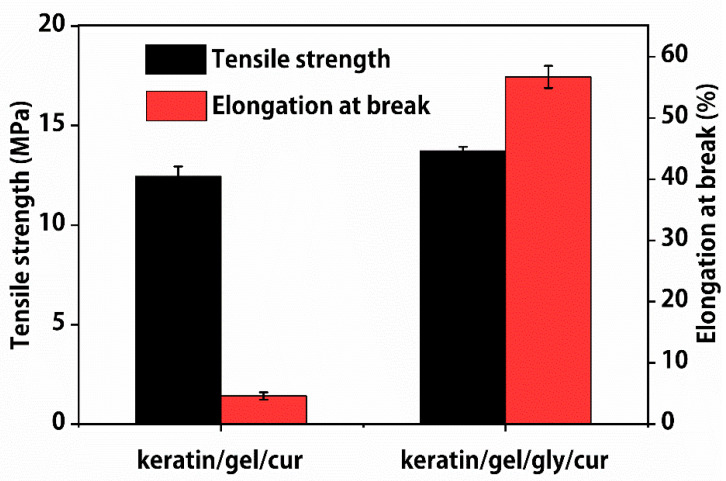
Tensile strength and elongation at break properties of keratin (7%)/gelatin (10%)/curcumin (1.0%) and keratin (7%)/gelatin (10%)/glycerin (2.0%)/curcumin (1.0%) composite films.

**Figure 5 molecules-27-06331-f005:**
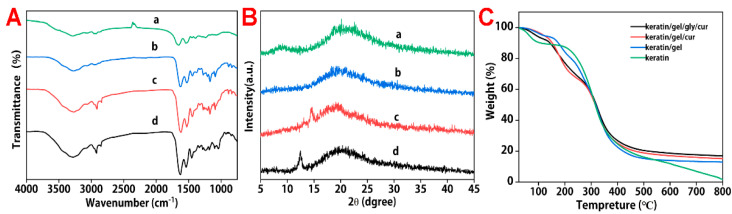
FTIR (**A**), XRD (**B**), and TGA (**C**) spectra of (**a**) pure keratin, (**b**) keratin/gelatin, (**c**) keratin/gelatin/ curcumin, and (**d**) keratin/gelatin/glycerin/curcumin composite films.

**Figure 6 molecules-27-06331-f006:**
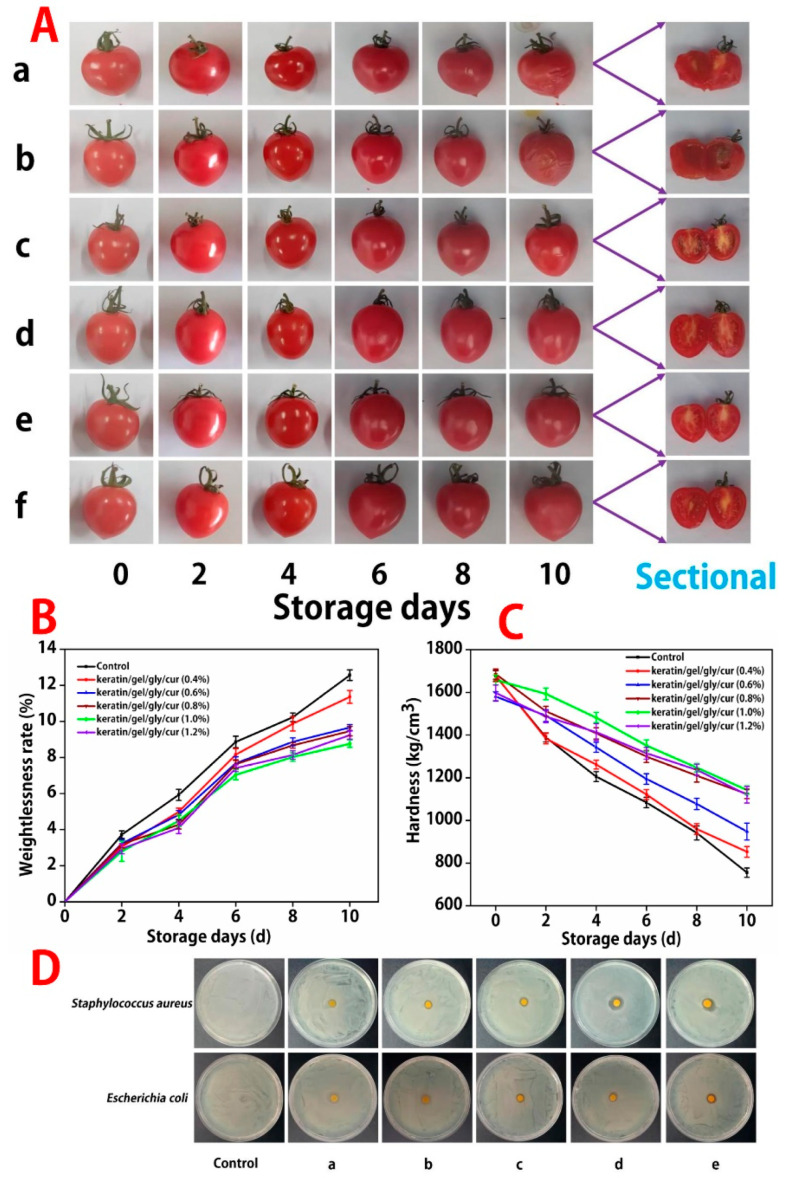
Images of (**A**) tomatoes treated with different keratin/gel/gly/cur composite films before and after storage for 10 days, (**a**): not film wrapped, (**b**–**f**) the concentrations of curcumin: 0.4, 0.6, 0.8, 1.0, and 1.2%, (**B**,**C**) The effects of different concentrations of curcumin on weight-loss rate and hardness before and after 10 days, and (**D**) the antibacterial property of different concentrations of curcumin for keratin/gel/gly/cur composite films on two pathogens (*Staphylococcus aureus* and *Escherichia coli*), control: no film placed, (**a**–**e**) the concentrations of curcumin: 0.4, 0.6, 0.8, 1.0, and 1.2%.

**Figure 7 molecules-27-06331-f007:**
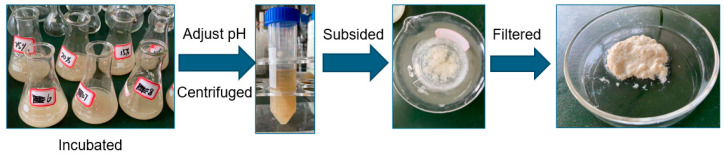
Schematic of the extraction of pig nail keratin.

**Table 1 molecules-27-06331-t001:** Effects of different concentration of curcumin on color attributes of cherry tomatoes stored at room temperature for 10 days.

Days		Control	1	2	3	4	5
Luminosity (L*)	0	30.25 (0.41) ^a,u^	30.07 (0.61) ^a,u^	30.27 (0.29) ^a,u^	30.92 (0.34) ^a,u^	30.03 (0.31) ^a,u^	30.62 (0.97) ^a,u^
	2	28.93 (0.36) ^b,u^	29.27 (0.19) ^b,u^	29.28 (0.89) ^a,b,u^	29.81 (0.14) ^b,u^	29.54 (0.13) ^a,b,u^	30.16 (0.53) ^a,b,u^
	4	28.26 (0.18) ^b,w^	28.38 (0.20) ^b,c,v,w^	28.67 (0.16) ^b,c,u,v,w^	28.85 (0.15) ^c,u,v^	28.63 (0.31) ^b,c,v,w^	29.20 (0.29) ^b,c,u^
	6	27.53 (0.17) ^c,v^	27.62 (0.25) ^c,u,v^	28.03 (0.55) ^c,d,u,v^	28.26 (0.26) ^d,u,v^	28.02 (0.48) ^c,d,u,v^	28.61 (0.53) ^b,c,d,u^
	8	26.82 (0.46) ^d,v^	27.06 (0.13) ^d,u,v^	27.26 (0.13) ^d,e,u,v^	27.68 (0.13) ^d,e,u,v^	27.60 (0.85) ^c,d,u,v^	28.19 (0.33) ^c,d,u^
	10	26.03 (0.18) ^e,w^	26.36 (0.28) ^d,v^	26.80 (0.12) ^e,v^	27.15 (0.34) ^e,u,v^	27.24 (0.49) ^d,u,v^	27.62 (0.19) ^d,u^
Chroma (C*ab)	0	28.94 (0.92) ^a,u,v,w^	28.98 (1.46) ^a,u,v,w^	26.92 (0.47) ^a,w^	27.20 (0.94) ^a,v,w^	29.71 (1.25) ^a,u,v^	31.05 (1.26) ^a,u^
	2	27.19 (0.52) ^a,v,w^	28.58 (0.94) ^a,b,u,v^	25.97 (0.24) ^a,b,w^	26.07 (0.77) ^a,b,w^	28.77 (0.64) ^a,b,u,v^	30.11 (0.83) ^a,u^
	4	26.98 (1.47) ^a,u,v^	26.88 (0.96) ^a,b,c,u,v^	25.72 (0.30) ^a,b,v^	25.34 (0.71) ^b,c,v^	27.33 (0.32) ^b,c,u,v^	29.04 (1.35) ^a,u^
	6	26.40 (1.46) ^a,u,v,w^	26.53 (0.28) ^b,c,u,v,w^	25.68 (1.21) ^a,b,v,w^	24.43 (0.43) ^b,c,w^	27.44 (0.50) ^b,c,u,v^	28.61 (0.50) ^a,u^
	8	22.75 (1.20) ^b,x^	25.84 (0.41) ^c,u,v,w^	24.40 (1.45) ^b,c,v,w,x^	23.96 (0.60) ^c,w,x^	27.26 (0.28) ^b,c,u,v^	28.28 (2.39) ^a,u^
	10	22.39 (0.50) ^b,x^	25.13 (1.26) ^c,v,w^	23.35 (1.10) ^c,w,x^	23.82 (0.71) ^c,v,w,x^	26.50 (0.90) ^c,u,v^	28.16 (1.83) ^a,u^
Hue angle (H*ab)	0	30.40 (1.66) ^a,u^	29.06 (1.36) ^a,u,v^	29.11 (0.81) ^a,u,v^	27.51 (0.72) ^a,v^	27.18 (0.26) ^a,v^	28.94 (0.27) ^a,u,v^
	2	30.30 (1.06) ^a,u^	28.67 (1.04) ^a,u^	28.95 (0.68) ^a,u^	26.01 (0.72) ^a,w^	26.59 (0.49) ^a,v,w^	28.36 (0.75) ^a,b,u,v^
	4	27.06 (2.44) ^ab,u^	27.00 (1.02) ^a,u^	27.17 (0.38) ^a,b,u^	25.96 (2.30) ^a,u^	26.09 (0.38) ^a,u^	27.90 (0.30) ^a,b,u^
	6	25.84 (1.59) ^b,u^	25.4 (0.86) ^a,b,u^	26.23 (1.58) ^b,u^	25.58 (0.74) ^a,u^	25.34 (0.92) ^a,u^	27.73 (0.91) ^a,b,u^
	8	22.43 (1.04) ^c,w^	20.94 (2.36) ^b,c,v,w^	25.32 (0.65) ^b,u^	24.09 (0.26) ^a,b,u,v^	24.86 (0.89) ^a,b,u^	26.68 (1.36) ^a,b,u^
	10	19.43 (0.80) ^c,v^	18.57 (3.71) ^c,v^	21.83 (1.61) ^c,u,v^	20.89 (3.35) ^b,u,v^	22.29 (2.40) ^b,u,v^	26.08 (1.60) ^b,u^

Mean values and standard deviation (in brackets); Different superscripts (^a–e^) within a column indicate significant differences during the storage ANOVA test (*p* < 0.05); Different superscripts (^u–x^) within a row indicate significant differences among packaging treatments according to the ANOVA test (*p* < 0.05); Different order number (1–5) indicate different keratin/gelatin/glycerin/curcumin composite films with different concentration of curcumin (0.4, 0.6, 0.8, 1.0, and 1.2%).

## Data Availability

Data are contained within the article.

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
