# Peer review of "Keratin-Based Composite Bioactive Films and Their Preservative Effects on Cherry Tomato"

_molecules, 2022, doi:10.3390/molecules27196331_

Round 1

Reviewer 1 Report (New Reviewer)

Dear Authors,

Different articles recently published regarding bioactive materials, curcumin, and food preservation must be added to the introduction part of the study. In this respect, some examples are given for the authors. 10.1016/j.lwt.2019.108292, https://doi.org/10.1111/jfpp.16538, also, the studies related to food preservation based on novel technology must be given in the introduction. Moreover, it is too short and must be enlarged.

SEM images should be improved as much possible as. 

Author Response

Response to Reviewer 1 Comments

Point 1: Different articles recently published regarding bioactive materials, curcumin, and food preservation must be added to the introduction part of the study. In this respect, some examples are given for the authors. 10.1016/j.lwt.2019.108292, https://doi.org/10.1111/jfpp.16538, also, the studies related to food preservation based on novel technology must be given in the introduction. Moreover, it is too short and must be enlarged.

Response 1: Thank you for your opinion, this part has been revised and the information has been added.

Point 2: SEM images should be improved as much possible as.

Response 2: Thank you for your opinion, this SEM images got from large instrument analysis platform in our school and no changes have been made. Maybe the resolution of the picture is not enough, we will consider changing the platform in the future research.

Reviewer 2 Report (New Reviewer)

In this work, the authors investigated the effect of Keratin based composite bioactive films on preservation of cherry tomato. This topic is interesting, and the film is also well characterized. However, I have some comments that need careful consideration from the authors.

the language should be carefully checked and improved. In many places, I find it difficult for me to understand. For example, the title can be remodeled as “Keratin-based Composite bioactive Films and their preservative effects on Cherry Tomato”.

Line 13-15, I cannot understand this sentence “Curcumin were successively incorporated in keratin, gelatin, glycerin to prepare several kinds of films included different concentration of keratin and curcumin composite films and the novel keratin (7.0%)/ gelatin (10%)/glycerin (2.0%)/curcumin (1.0%) composite film has the best effect on the tomatoes preservation.” Please rewrite it in a clearer way.

Line 23, what is “the weightlessness rate”. Please avoid Chinglish.

Line 25-26. You mentioned that “The inhibition effect of keratin/gelatin/glycerin/curcumin (1.0%) on S. aureus (Staphylococcus aureus) and E. coli (Escherichia coli) was investigated.” But how is the inhibitory effect?

Abstract should be highly summarized rather than only telling people what you did.

Line 32, You state “Cherry tomato is a source of nutrients and other healthy minerals that benefit the human body”. What nutrients does tomato contain? Please clarify.

Line 96, section 2.3. I could not find how curcumin was added. Please check.

Line 179-191, how did you define the β-sheet structure from the XRD-pattern? Please find a citation to support your statement.

Another comment is that, although the keratin-based films are well characterized, I would like to see a comparison between the film prepared in your study and the films prepared in other’ works, from which the advantages of the keratin-based films may be highlighted.

Author Response

Response to Reviewer 2 Comments

In this work, the authors investigated the effect of Keratin based composite bioactive films on preservation of cherry tomato. This topic is interesting, and the film is also well characterized. However, I have some comments that need careful consideration from the authors.

Point 1: the language should be carefully checked and improved. In many places, I find it difficult for me to understand. For example, the title can be remodeled as “Keratin-based Composite bioactive Films and their preservative effects on Cherry Tomato”.

Response 1: Thank you for your opinion, the title has been revised.

Point 2: Line 13-15, I cannot understand this sentence “Curcumin were successively incorporated in keratin, gelatin, glycerin to prepare several kinds of films included different concentration of keratin and curcumin composite films and the novel keratin (7.0%)/ gelatin (10%)/glycerin (2.0%)/curcumin (1.0%) composite film has the best effect on the tomatoes preservation.” Please rewrite it in a clearer way.

Response 2: Thank you for your opinion, this sentence has been revised in the abstract.

Point 3: Line 23, what is “the weightlessness rate”. Please avoid Chinglish.

Response 3: Thank you for your opinion, “the weightlessness rate” has been deleted, “the ratio of weight loss” has been added.

Point 4: Line 25-26. You mentioned that “The inhibition effect of keratin/gelatin/glycerin/curcumin (1.0%) on S. aureus (Staphylococcus aureus) and E. coli (Escherichia coli) was investigated.” But how is the inhibitory effect?

Response 4: Thank you for your opinion, the information has been added in the abstract.

Point 5: Abstract should be highly summarized rather than only telling people what you did.

Response 5: Thank you for your opinion, the abstract has been summarized.

Point 6: Line 32, You state “Cherry tomato is a source of nutrients and other healthy minerals that benefit the human body”. What nutrients does tomato contain? Please clarify.

Response 6: Thank you for your opinion, the information has been added in the introduction.

Point 7: Line 96, section 2.3. I could not find how curcumin was added. Please check.

Response 7: Thank you for your opinion, the addition process of curcumin has been explained clearly.

Point 8: Line 179-191, how did you define the β-sheet structure from the XRD-pattern? Please find a citation to support your statement.

Response 8: Thank you for your opinion, this part has been revised and the reference has been added.

Point 9: Another comment is that, although the keratin-based films are well characterized, I would like to see a comparison between the film prepared in your study and the films prepared in other’ works, from which the advantages of the keratin-based films may be highlighted.

Response 9: Thank you for your opinion, in this manucript keratin-based films are characterized, because of the addition of curcumin, the composite films have bacteriostasis performance. However, at present, the fresh-keeping film sold in the market do not have this property, we do not make comparison between them. This is a good advice, we will think about how to carry out comparative experiments in the future research, thank you very much.

Reviewer 3 Report (New Reviewer)

The present manuscript deals with the formulation of keratin based composite films containing curcumin and intended for prolonging the shelf-life of cherry tomatoes.

Introduction:

Line 33-34 Why should  tomatoes be sensitive than other fruits and vegetables to storage enviroment and trasportation? Please revise the sentence.

Some references in the manuscript are not consistent with the sentence referring to. For instance, Reference [6] is  about chitosan biodegradable films and not non degrable petroleum based packaging films. Reference [22] does not report the keratin extraction procedure. Please revise accurately all references cited in the manuscript if they are appropriated.

Line 61 Gelatin as excipient used for the preparation of the film is not introduced in a readable manner inside the introduction. The sentence “Gelatin film is limited by poor antioxidant activity” is unmeaningful in the context in which it appears.

Moreover, It is not well defined the role of gelatin in preparing such film and the reason of chosing curcumin as active ingredient in the formulation of film, especially regarding the formulation of packaging film. Therefore, the rationale and aim of the study at the end of the introduction should be revised (lines 61-73).

Materials and methods:

Line 78 The Bloom grade is not indicated by letter “g”. Please remove “g”.

Please indicate the unit of the molecular mass as Dalton.

Line 90 In which sense “ the supernatant was extracted”?

Please add an appropriate reference for the extraction procedure. Was the yield of the extraction calculated? Which is the range of molecular weights of the extracted keratin?

Paragraph 2.3 The paragraph is confusing and it is hard to understand which formulations were prepared and at which concentrations each component was employed. Please revise all paragraph.

Why hexafluoroisopropanol was used as solvent?

Line 105 Does blast dryer mean a drying oven?

Which volume of each solution was poured into the mold to prepare the films?

Paragraph 2.4 Was the thickness of the film measured? The thickness is important to evaluate the meachanical properties.

Line 110 ATR-IR spectroscopy did not confirm the molecular structure of nanofiber membranes.

How tensile strenght and elongation at break were calculated from raw data?

Paragraph 2.6 How were tomatoes wrapped up with the composite films? Have films the required flexibility?

Paragraph 2.9  Is hardeness a destructive test in comparison to the other performed to evaluate tomatoes preservation?

Results and discussion

Paragraph 3.1 It is not clear to what pig nails powder and pig nails keratin refer to. All paragraph must be revised and implemented.

Paragraph 3.2 It is not clear to what pig nails powder and pig nails keratin refer to. Keratin is not a crystalline powder. I think it is not correct to write “change in cristallinity” in line 176. All paragraph must be revised and implemented.

Paragraph 3.3. It is not clear which is the exact composition of the analysed films. It is not clear which are the concentration used for the casting dispersion poured in the mold and the effective concentration of each component in the film. Moreover, results obtained from different percentages of keratin are shown, but it is unknown which are and how the final percentages of gelatin and glycerol were selected. All this paragraph must be revised together with the paragraph reporting the composite films preparation in the method section.

Line 195 Why pectin? I guess pectin was not employed for the preparation of films.

Paragraph 3.4 Also here, it is not clear the composition of the prepared films. Line 212-214 this sentence is repetitive since elongation at break result was already explained above.

Line 209-212 Please explain better the effect of glycerol on the tensile strenght and elongation at break of films.

WHich is the effect of the curcumin incorporation on mechanical properties?

Was the mechanical properties of the films stable over time, at least for ten days, which is the time for tomatoes preservation evaluation assays? How much time after preparation were films applied onto tomatoes?

Paragraph 3.5

Lines 228-229 Please explain better the relevance of the results obtained from FT-IR spectra.

Line 230-235 To assess any variation in a certain state of cristallinity, gelatin and curcumin alone should be also analysed by XRD.

Line 237-238 In which sense results from XRD were inconsistent with SEM observations?

Line 246-248 The sentence is not clear since there is confusion between the second and third step of degradation. Please rephase and explain better the scientific meanng of the degradation steps in the composites films.

Paragraph 3.6

Line 264-265 How was determined that keratin/gel/gly/cur (1.0%) composite film had a better preservation?

Are all concentration percentages w/v in all manuscript?

Figure 7D there is a typing mistake for Staphilococcus aureus.

Please consider if to make changes in the conclusion and abstract after the revision of the other sections.

Author Response

Response to Reviewer 2 Comments

The present manuscript deals with the formulation of keratin based composite films containing curcumin and intended for prolonging the shelf-life of cherry tomatoes.

Point 1: Line 33-34 Why should tomatoes be sensitive than other fruits and vegetables to storage enviroment and trasportation? Please revise the sentence.

Response 1: Thank you for your opinion, the sentence has been revised in the introduction.

Point 2: Some references in the manuscript are not consistent with the sentence referring to. For instance, Reference [6] is about chitosan biodegradable films and not non degrable petroleum based packaging films. Reference [22] does not report the keratin extraction procedure. Please revise accurately all references cited in the manuscript if they are appropriated.

Response 2: Thank you for your opinion, reference [6] and [22] have been revised.

Point 3: Line 61 Gelatin as excipient used for the preparation of the film is not introduced in a readable manner inside the introduction. The sentence “Gelatin film is limited by poor antioxidant activity” is unmeaningful in the context in which it appears.

Response 3: Thank you for your opinion, the sentence in the introduction has been deleted.

Point 4: Moreover, It is not well defined the role of gelatin in preparing such film and the reason of chosing curcumin as active ingredient in the formulation of film, especially regarding the formulation of packaging film. Therefore, the rationale and aim of the study at the end of the introduction should be revised (lines 61-73).

Response 4: Thank you for your opinion, the bacteriostatic activity of curcumin and the role of gelatin we got from reference. And in the course of the experiment, it has found that the addition of glycerol can make the film have good softness. In addition, the introduction has been revised follow my understanding.

Point 5: Line 78 The Bloom grade is not indicated by letter “g”. Please remove “g”.

Response 5: Thank you for your opinion, “g” has been removed.

Point 6: Please indicate the unit of the molecular mass as Dalton.

Response 6: Thank you for your opinion, the Dalton has been added.

Point 7: Line 90 In which sense “ the supernatant was extracted”?

Response 7: Thank you for your opinion, it’s my fault, “the supernatant was collected” has been added.

Point 8: Please add an appropriate reference for the extraction procedure. Was the yield of the extraction calculated? Which is the range of molecular weights of the extracted keratin?

Response 8: Thank you for your opinion, the results show that: The optimal extraction conditions of porcine hoof nail keratin were as follows: the amount of L-cysteine was 15% (the amount of L-cysteine accounted for the mass of nail powder), the concentration of urea solution was 4 mol/L, the reaction pH was 11.5, the reaction temperature was 70℃, the reaction time was 16 h, and the crushing method of porcine hoof nail was machine crushing. Under these conditions, the extraction rate of porcine hoof beetle keratin was 81%.

The ration of extraction=( M2/M1 ) × 100%

where M1 represents the initial weight of the pig nail before exraction and M2 represents the weight of the pig nail keratin.

Figure 3. The effect of L-cystenine concentration, reaction time, pH, temperature, urea solution concentration and grinding method on the extraction efficiency.

Point 9: Line 110 ATR-IR spectroscopy did not confirm the molecular structure of nanofiber membranes.

Response 9: Thank you for your opinion, the “the molecular structure”has been replaced by“functional groups”.

Point 10: Paragraph 2.6 How were tomatoes wrapped up with the composite films? Have films the required flexibility?

Response 10: Thank you for your opinion, the films was flexibe and can wrap tomatoes.

Point 11: Paragraph 2.9 Is hardeness a destructive test in comparison to the other performed to evaluate tomatoes preservation?

Response 11: Thank you for your opinion, compared with other indicators, hardeness was measured by pressing the tomato with a needle until a hole appears on the surface.

Point 12: Why hexafluoroisopropanol was used as solvent?

Response 12: Thank you for your opinion, the pig nail powders can be fully dissolved in hexafluoroisopropanol.

Point 13: Line 105 Does blast dryer mean a drying oven?

Response 13: Thank you for your opinion, it’s drying oven and it has been revised.

Point 14: Which volume of each solution was poured into the mold to prepare the films?

Response 14: Thank you for your opinion, the volume was about 5mL.

Point 15: Paragraph 2.4 Was the thickness of the film measured? The thickness is important to evaluate the meachanical properties.

Response 15: Thank you for your opinion, the thickness of the film was about 0.04-0.05mm. Because the film is very thin, the thickness of the film was obtained by folding the film for 3 layers and then measured with a vernier caliper, the final number is divided by 8, shown in this figure.

Point 16: Paragraph 3.1 It is not clear to what pig nails powder and pig nails keratin refer to. All paragraph must be revised and implemented.

Response 16: Thank you for your opinion, pig nails keratin was extracted from pig nails powder, but the pig nails powder just pretreated by only ground. In order to be distinguished, the pig nails powder and pig nails keratin were used in the paragraph.

Point 17: Paragraph 3.2 It is not clear to what pig nails powder and pig nails keratin refer to. Keratin is not a crystalline powder. I think it is not correct to write “change in cristallinity” in line 176. All paragraph must be revised and implemented.

Response 17: Thank you for your opinion, this part has been revised in the manucript.

Point 18: Paragraph 3.3. It is not clear which is the exact composition of the analysed films. It is not clear which are the concentration used for the casting dispersion poured in the mold and the effective concentration of each component in the film. Moreover, results obtained from different percentages of keratin are shown, but it is unknown which are and how the final percentages of gelatin and glycerol were selected. All this paragraph must be revised together with the paragraph reporting the composite films preparation in the method section.

Response 18: Thank you for your opinion, the information of the concentration for films has been added in the manucript.

Point 19: Line 195 Why pectin? I guess pectin was not employed for the preparation of films.

Response 19: Thank you for your opinion, it’s my fault it has been deleted, and the gelatin has been added.

Point 20: Paragraph 3.4 Also here, it is not clear the composition of the prepared films. Line 212-214 this sentence is repetitive since elongation at break result was already explained above.

Response 20: Thank you for your opinion, the information about the composition of the prepared films was added in the manucript; and the repetitive sentence has been deleted.

Point 21: How tensile strenght and elongation at break were calculated from raw data?

Response 21: Thank you for your opinion, this experiment was implemented by the company and the data was got from comppany. However, we will study how the research data is calculated.

Point 22: Line 209-212 Please explain better the effect of glycerol on the tensile strenght and elongation at break of films.

Response 22: Thank you for your opinion, the information has been added.

Point 23: WHich is the effect of the curcumin incorporation on mechanical properties?

Response 23: Thank you for your opinion, curcumin was used as bacteriostatic, there is no information about the the effect of mechanical properties. However, this advice is very important and we will focus on this aspect.

Point 24: Lines 228-229 Please explain better the relevance of the results obtained from FT-IR spectra.

Response 24: Thank you for your opinion, this part has been revised.

Point 25: Was the mechanical properties of the films stable over time, at least for ten days, which is the time for tomatoes preservation evaluation assays? How much time after preparation were films applied onto tomatoes?

Response 25: Thank you for your opinion, Eventhough, the mechanical properties changed with time have not been studied, the tomatoes wrapped up with the composite films follow this figure, and after ten days, the films were not cracked. The films were applied onto tomatoes for about ten days.

Point 26: Line 230-235 To assess any variation in a certain state of cristallinity, gelatin and curcumin alone should be also analysed by XRD.

Response 26: Thank you for your opinion, pure keratin, keratin/gelatin, keratin/gelatin/ curcumin and keratin/gelatin/glycerin/curcumin have been shown the different composition of the films. And we also get the change from the addition of gelatin and curcumin by comparing keratin and keratin/gelatin, and the keratin/gelatin and keratin/gelatin/ curcumin.

Point 27: Line 237-238 In which sense results from XRD were inconsistent with SEM observations?

Response 27: Thank you for your opinion, the sentence has been deleted.

Point 28: Line 246-248 The sentence is not clear since there is confusion between the second and third step of degradation. Please rephase and explain better the scientific meanng of the degradation steps in the composites films.

Response 28: Thank you for your opinion, this part has been revised.

Point 29: Line 264-265 How was determined that keratin/gel/gly/cur (1.0%) composite film had a better preservation?

Response 29: Thank you for your opinion, the determination of the keratin/gel/gly/cur (1.0%) composite film had better preservation by using the indicators included antibacterial property of composite films, the ratio of weight loss, hardness and surface color in the manucript.

Point 30: Are all concentration percentages w/v in all manuscript?

Response 30: Thank you for your opinion, all concentration percentages are w/v in all manuscript and the information has been added in the Paragraph 2.3.

Point 31: Figure 7D there is a typing mistake for Staphilococcus aureus.

Response 31: Thank you for your opinion, it is my fault, and it has been revised, than you very much.

Point 32: Please consider if to make changes in the conclusion and abstract after the revision of the other sections.

Response 32: Thank you for your opinion, these two parts have been revised.

Round 2

Reviewer 1 Report (New Reviewer)

Ok 

Author Response

No comments

Reviewer 3 Report (New Reviewer)

Response 11: Thank you for your opinion, compared with other indicators, hardeness was measured by pressing the tomato with a needle until a hole appears on the surface.

Reviewer: It is not still clear how the hardness was measured. Is it used a needle or directly using the plunger of the texture analyzer? More information about the method used must be added in the manuscript.

Response 15: Thank you for your opinion, the thickness of the film was about 0.04-0.05mm. Because the film is very thin, the thickness of the film was obtained by folding the film for 3 layers and then measured with a vernier caliper, the final number is divided by 8, shown in this figure.

The thickness of the films is not reported in the manuscript. The information about the methoud used to measure thickness of films and the obtained results must be added in the manuscript.

Response 21: Thank you for your opinion, this experiment was implemented by the company and the data was got from company. However, we will study how the research data is calculated.

I understand, but the method used must be exhaustive and reproducible in a scientific manuscript. The equations used to calculate tensile strenght and elongation at break must be reported.

Author Response

Reviewer: It is not still clear how the hardness was measured. Is it used a needle or directly using the plunger of the texture analyzer? More information about the method used must be added in the manuscript.

Response 32: Thank you for your opinion, the addition has been added in the section 2.9. And the probe was shown in the picture.

Reviewer: The thickness of the films is not reported in the manuscript. The information about the method used to measure thickness of films and the obtained results must be added in the manuscript.

Response: Thank you for your opinion, the addition has been added in the section 2.10, and shown in the figure.

Reviewer: I understand, but the method used must be exhaustive and reproducible in a scientific manuscript. The equations used to calculate tensile strengh and elongation at break must be reported.

Response: Thank you for your opinion, the addition has been added in the section 2.

This manuscript is a resubmission of an earlier submission. The following is a list of the peer review reports and author responses from that submission.

Round 1

Reviewer 1 Report

In this manuscript, authors developed composite film based on pig nail extracted keratin modified with gelatin and glycerin. The manuscript was written well. Following are my queries and suggestion:

·       Authors extracted keratin from pig nails and formulated composite film using keratin, gelatin, Glycerin and curcumin. And compared the composite film of keratin+gelatin+curcumin (formulation 1) and keratin+gelatin+glycerin+curcumin (formulation 2). The novelty of this work is the application of keratin extracted from pig nail waste, so it would be ideal to compare the film developed with commercial keratin and with pig nail keratin.

·        Line 49: Authors can also provide the information of yield of protein content extracted from pig nails.

·        The purpose of selecting cherry tomatoes for this study can be given in brief in the introduction section.

·        Section 2: From Table 1 captions, I believe authors conducted ANOVA test. But the description of statistical analysis is missing in “Materials and method” section.

·        Section 3.4: Line 200: Authors mentioned addition of glycerol in Keratin/gelatin/glycerol/curcurmin composite films improved the elongation at break. Further in line 205 it was mentioned as the elongation at break was 56.7 %. However, in figure 5, it shows less than 5 %.

·        Authors conducted antibacterial study for the composite film. But discussing in section 3.6 resembles the antibacterial study was conducted for chery tomatoes coated with composite films. Authors can discuss the antibacterial study in a separate section.

·        Line 274: “Weight loss rate of cherry tomatoes” were appearing suddenly, and the concept of weight loss was not discussed before.

·        In section 3.6. and 3.8, authors compared the coated cherry tomatoes with control. However “control” is not defined anywhere in the text. Does the authors mean “control” as tomatoes without any coating. 

Reviewer 2 Report

This paper has unacceptable parts related to the microbiological aspect of the work. Starting with misspelled names of microorganisms, unexplained and confused methodologies for antimicrobial testing with monoculture or co-culture!? and interpretation of the results of these tests.

I would like to ask that the entire text be revised in terms of a significant improvement in the microbiological aspect because unacceptable errors were observed. Also, the obtained results are not described and shown in the right manner. Please, check some scientific-relevant papers to see how to do this.